# The Lavic Lake Fault: A Long-Term Cumulative Slip Analysis via Combined Field Work and Thermal Infrared Hyperspectral Airborne Remote Sensing

Rebecca A. Witkosky [1,*], Joann M. Stock [1], David M. Tratt [2], Kerry N. Buckland [2], Paul M. Adams [2], Patrick D. Johnson [2], David K. Lynch [3] and Francis J. Sousa [4]

[1] California Institute of Technology, Division of Geological and Planetary Sciences, 1200 E. California Blvd., MC 170-25, Pasadena, CA 91125, USA; jstock@gps.caltech.edu

[2] The Aerospace Corporation, 2310 E. El Segundo Blvd., El Segundo, CA 90245, USA; david.m.tratt@aero.org (D.M.T.); Kerry.N.Buckland@aero.org (K.N.B.); paul.m.adams@aero.org (P.M.A.); Patrick.D.Johnson@aero.org (P.D.J.)

[3] Thule Scientific, P.O. Box 953, Topanga, CA 90290, USA; dave@caltech.edu

[4] College of Earth, Ocean, and Atmospheric Sciences, Oregon State University, 104 CEOAS Administration Building, Corvallis, OR 97331-5503, USA; francis.sousa@oregonstate.edu

\* Correspondence: amber@alumni.caltech.edu

**Abstract:** The 1999 Hector Mine earthquake ruptured to the surface in eastern California, with >5 m peak right-lateral slip on the Lavic Lake fault. The cumulative offset and geologic slip rate of this fault are not well defined, which inhibits tectonic reconstructions and risk assessment of the Eastern California Shear Zone (ECSZ). With thermal infrared hyperspectral airborne imagery, field data, and auxiliary information from legacy geologic maps, we created lithologic maps of the area using supervised and unsupervised classifications of the remote sensing imagery. We optimized a data processing sequence for supervised classifications, resulting in lithologic maps over a test area with an overall accuracy of 71 ± 1% with respect to ground-truth geologic mapping. Using all of the data and maps, we identified offset bedrock features that yield piercing points along the main Lavic Lake fault and indicate a 1036 +27/−26 m net slip, with 1008 +14/−17 m horizontal and 241 +51/−47 m vertical components. For the contribution from distributed shear, modern off-fault deformation values from another study imply a larger horizontal slip component of 1276 +18/−22 m. Within the constraints, we estimate a geologic slip rate of <4 mm/yr, which does not increase the sum geologic Mojave ECSZ rate to current geodetic values. Our result supports previous suggestions that transient tectonic activity in this area may be responsible for the discrepancy between long-term geologic and present-day geodetic rates.

**Keywords:** hector mine earthquake; lavic lake fault; thermal infrared hyperspectral airborne imagery, net slip, eastern california shear zone

## 1. Introduction

The 1992 $M_w$ 7.3 Landers and 1999 $M_w$ 7.1 Hector Mine earthquakes resulted in two major surface ruptures in eastern California (Figure 1). The faults that ruptured in the 1992 Landers event were mostly located on publicly accessible land, allowing considerable subsequent geologic investigation [1–9]. However, faults that ruptured in the 1999 Hector Mine event, which include the Lavic Lake fault, and portions of the Calico-Hidalgo, Mesquite Lake, and Bullion faults (Figure 2a), are located within the United States Marine Corps Air Ground Combat Center (MCAGCC, Twentynine Palms), and therefore received only limited field-based study [10].

The Lavic Lake fault is located in the Bullion Mountains of the Mojave Desert, but there is limited information on the geology of this area, and the long-term cumulative offset along the fault is not well resolved. In older maps, the Lavic Lake fault had not been formally named, its sense of displacement was unknown, and much of its surface trace was only mapped as an approximate location [14–19]. After the 1999 earthquake and surface rupture, Treiman et al. [10] were able to map the surface trace and determine that the fault slip was generally right lateral. The Hector Mine Earthquake Geologic Working Group [13] formally named the fault after the Lavic Lake Playa (Figure 2a). A cumulative fault offset measurement was then determined via geophysical methods: Jachens et al. [20] estimated 3400 ± 800 m of right-lateral offset from displaced magnetic anomaly pairs within the Bullion Mountains. However, the ages of the offset magnetic anomaly pairs are unknown, so a geologic fault slip rate could not be calculated and remains unknown. The geologic slip rate would be useful because of the discrepancy between integrated geologic fault slip rates and current geodetic crustal motion (~6 and 12 mm/yr, respectively) in eastern California. Estimates of long-term average geologic slip rates for the Eastern California Shear Zone (ECSZ) range from 8.3 ± 1.0 mm/yr since 12 Ma [21], to ≤6.2 ± 1.9 mm/yr since ~750 ka [22]. The discrepancy between geologic and geodetic slip rates in the ECSZ could be minimized by incorporating the geologic slip rates of additional active faults.

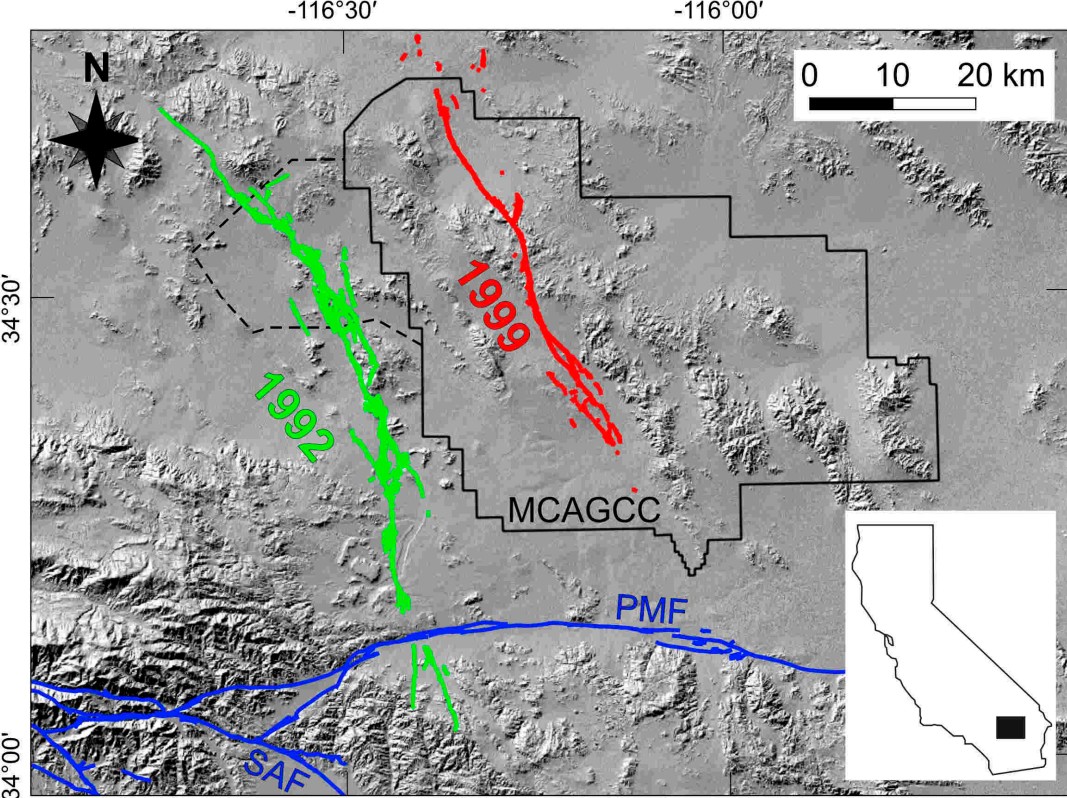

**Figure 1.** Overview map of the 1992 $M_w$ 7.3 Landers and 1999 $M_w$ 7.1 Hector Mine earthquake surface ruptures, and territory with access restricted by the MCAGCC (United States Marine Corps Air Ground Combat Center, Twentynine Palms) as of 1999 (solid line), and as of 2018 (dashed line). Other major faults shown are the Pinto Mountain fault (PMF) and the Mission Creek strand of the San Andreas fault (SAF). Base map is an SRTM 1-arcsecond DEM hill shade (from the USGS Earth Resources Observation and Science website, accessed September 2017 from https://eros.usgs.gov/). Faults are from [11]. Map was produced using [12].

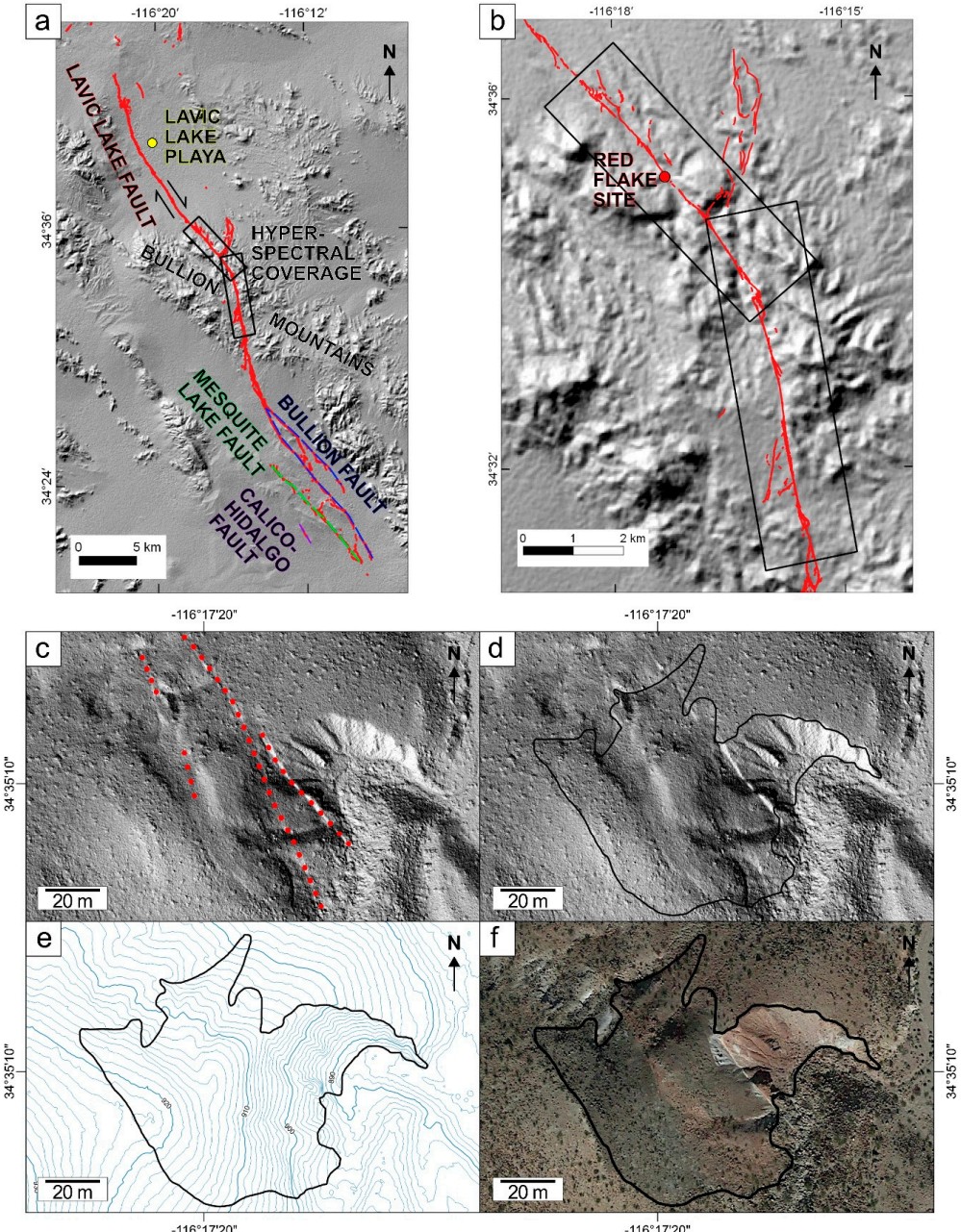

**Figure 2.** (**a**) Overview of the Lavic Lake fault, mapped by its 1999 earthquake surface rupture. Arrows indicate relative fault motion, and rectangular boxes are the extent of ground coverage of the hyperspectral airborne imagery used in this study. The Lavic Lake playa (after which the fault was named by [13]), and the Bullion Mountains, are labeled. GIS data (base map and faults) are same as in Figure 1. Portions of other faults that ruptured in the 1999 earthquake are annotated with dashed lines in distinct colors: Calico-Hidalgo fault zone, Mesquite Lake fault, and Bullion fault zone. (**b**) Overview of the portion of the Lavic Lake Fault covered by the hyperspectral airborne imagery footprints. The Red Flake site is an area that we ground-truthed for supervised classifications of the remote sensing imagery. GIS data are same as in Figure 1. (**c**–**f**) Overview maps of the Red Flake site. (**c**) Lidar 10-cm DEM hill shade (from the Open Topography website, accessed September 2017 from http://www.opentopography.org/). Fault scarps from the 1999 earthquake are annotated with dotted lines. (**d**) Polygon representing the ground truth area. Base map same as (**c**). (**e**) Elevation contour map (contour interval is 1 m). Polygon same as in (**d**). (**f**) Google Earth true color satellite image (imagery date: 2 January 2015). Polygon same as in (**d**). Maps in (**a**–**e**) were produced using [12].

Remote sensing methods can yield additional useful data for studying these faults. The coseismic slip of the 1999 Hector Mine earthquake has been investigated with several remote sensing methods: InSAR from [23,24]; InSAR and GPS from [25]; lidar-based offset measurements from [26,27]; lidar-based and field-based offset measurements from [28]. The remote sensing results are in general agreement with the coseismic slip measurements that were made in the field immediately following the 1999 earthquake by [10].

We previously used thermal infrared hyperspectral airborne imagery to produce a geologic swath map of the Lavic Lake fault [29]. With the hyperspectral airborne imagery, it was possible to differentiate various minerals and lithology within a scene by performing supervised and unsupervised classifications. These are algorithmic classification methods for grouping pixels in an image by quantifying and comparing spectral similarity. We first used supervised classifications, which required using lithologic ground truth from field mapping over a small area to quantify the accuracy of our image-processing sequence. We then took a hybrid approach to include advantages gained by mapping a larger area with unsupervised classifications. This allowed us to use our complete imagery dataset more effectively for remote geologic mapping over a greater spatial extent. Now, we combine those preliminary results with additional field observations, more remote sensing imagery products, and data from other geologic maps to estimate the long-term cumulative fault slip, which can be combined with bedrock ages to calculate a minimum geologic slip rate, and can also be used for palinspastic reconstructions of eastern California [21].

## 2. Geologic Setting

The Lavic Lake and Bullion faults are located in the Bullion Mountains of the Mojave Desert, California. The Bullion Mountains have 200–400 meters of topographic relief in bedrock outcrops. These include extrusive Tertiary and Quaternary igneous rocks, and underlying hypabyssal and intrusive lithologies of pre-Tertiary age [14]. Surficial weathering of the volcanic bedrock has resulted in alteration to abundant clay and zeolite minerals, and erosion has created Quaternary surficial deposits of colluvium and fanglomerate. In the northwest portion of the Bullion Mountains where Miocene to Oligocene igneous extrusive rocks crop out, the 1999 Hector Mine earthquake surface rupture along the Lavic Lake fault (Figure 2) reached a maximum right-lateral co-sesimic displacement exceeding 5 m [10]. Lithologic composition of these rocks ranges from andesitic to basaltic, with massive and porphyritic textures [15]. Bedding is upright in this area, and attitudes generally strike northwest with ≤30° dip to the northeast. Many small faults are present, striking approximately northwest with right-lateral separation; there are also faults that strike generally east–west, with either left-lateral or unknown sense of motion [15]. The other major structure in the area, the Bullion fault, strikes northwest and bounds the southwestern edge of the Bullion Mountains. This fault is sometimes combined with others to define the Bullion-Rodman-Pisgah fault (e.g., [30]) with net slip estimates that range from 6.4–14.4 km [31] up to 20–40 km [32].

Immediately north of the Bullion Mountains lies the Lavic Lake playa, which is composed mainly of hard packed clay [14], and contains decameter-scale surficial fractures and fissures that are linear and polygonal in shape. Farther north, the 22.5 ± 1.3 ka Pisgah basalts [33] are in contact with Lavic Lake playa deposits. Paleoseismic trenching and dating of Lavic Lake playa strata [34] show that the shallowest sediments postdate the Pisgah lavas. Nevertheless, because the thicknesses of these units are unknown, an interfingering contact between these two units cannot be ruled out. Paleoseismology on the Lavic Lake fault in the playa area following the 1999 earthquake yielded several important results. The surface trace of the 1999 event had not previously ruptured for at least ~7000 years, but another strand with geomorphic evidence (vegetation lineaments and uplifted basalt exposures) for recent activity ruptured sometime within the past ~1750 years. Rymer et al. [34] concluded that deformation has not yet been fully localized onto a single strand, suggesting that the Lavic Lake fault is relatively young. However, even if the fault is young, it is still possible that a significant amount of slip has accumulated over the course of multiple prehistoric earthquakes.

Regional deformation in eastern California transitioned from extensional to right-lateral shear from 6–8 Ma [35]. The Lavic Lake fault is one of many faults comprising a tectonic province formally named the Eastern California Shear Zone (ECSZ) by [36,37]. Estimates for the inception age of ECSZ faults vary, but generally fall within the range 5–10 Ma (Table 1). A minimum age for the onset of an ECSZ fault is provided by a 3.77 ± 0.11 Ma basaltic lava that drapes over a fault scarp in the Black Mountains, near the Garlock fault [38].

**Table 1.** Eastern California Shear Zone (ECSZ) age of inception.

| Author(s) | Age | Basis of Reasoning |
|---|---|---|
| Dokka and Travis [36,37] | 6–10 Ma | Initiation of Garlock Fault ~10 Ma [39], which is cut by younger ECSZ faults in the east [40,41]; age relations from [42], which may indicate that some ECSZ faults initiated c. 6 Ma; Paleomagnetic data from [43], which may indicate that regional deformation began after ~6 Ma |
| Schermer et al. [44] | <11.7 Ma | <11.7 Ma fan deposits and their older substrate are displaced the same amount by left-lateral faults in the northeastern Mojave Desert |
| Miller and Yount [45] | >5–6 Ma | East–west striking left-lateral ECSZ faults controlled topography and subsequently the flow direction of 5–6 Ma basalts |
| Gan et al. [46] | 5.0 ± 0.4 Ma | Modeling the deflection of the Garlock Fault's once straight, but now curved surface trace |
| Oskin and Iriondo [38] | >3.77 ± 0.11 Ma | Dated basalt flow that drapes a fault scarp in the Black Mountains |
| McQuarrie and Wernicke [21] | ~12 Ma | Right lateral shear, oriented N25° W since ~12 Ma is based on palinspastic restoration modeling of mountain ranges in the southwestern U.S.A. |
| Woodburne [47] | ~6 Ma | Coupled with, or possibly as a byproduct of the opening of the Gulf of California [48–51]; also cites a period of non-deposition in the Mojave Desert Region until ~6 Ma to argue for tectonic quiescence before that time |

In compiling the tectonic history of the ECSZ, the addition of net slip across all faults can be combined with age of inception to infer long-term geologic slip rates across the region. Earlier estimates across all northwest-striking, right-lateral faults striking generally varied between 25 and 65 km [30,31,36,52], but more recent estimates increase to about 100 km [21,53]. The models used to derive net slip typically invoke clockwise rotation of fault blocks [21,30], as this also can explain left-lateral slip along east–west striking faults within the ECSZ (e.g., see the model of [54]). However, some opposing models imply counterclockwise rotation of the fault blocks bounded by right-lateral faults [32,36]. While some studies [55,56] show compelling paleomagnetic evidence for early Miocene clockwise rotation in the Mojave Desert, MacFadden et al. [57,58] also pointed out that paleomagnetic results can vary locally and temporally, showing that caution is required in invoking a single generalized model of uniform rotation across the entire region.

Tallied net slips have been combined with age of inception to calculate the long-term geologic slip rate across the entire region, but results vary widely from 3 to 12 mm/yr since the Early, Middle, or Late Miocene [37]. A more recent and precise approach, however, involves extensive field work to find dateable offset piercing points along as many of the active faults in the region as possible, then integrating these results into a single value. Oskin et al. [22] took that approach, combining Quaternary offsets from six major faults to determine a "sum geologic Mojave ECSZ slip rate" of ≤6.2 ± 1.9 mm/yr since ~750 ka. The inequality marker indicates that they used maximum allowable

offset values (and subsequent rates) for six specific faults to derive the summed rate. On the other hand, the summed rate also represents a minimum, since data do not exist for every single active fault strand within the area of the integration, and the rate also does not account for off-fault strain (e.g., [21]).

The sum geologic rate of [22] highlights a discrepancy between results from geology and geodesy, despite the joint approach often used to study earthquakes in southern California. With GPS tracking of tectonic motion, geodetic slip rate estimates across the ECSZ are usually >10 mm/yr [59–65], much faster than the ≤6.2 ± 1.9 mm/yr sum geologic rate. Some geodetic slip rates are <10 mm/yr, but they are either based on older data [66,67], alternative methods [68], or are modeled with greatest effort to agree with the sum geologic rate and thus resolve the discrepancy (e.g., [69]). Meade and Hager [70], Oskin et al. [22], and Spinler et al. [64] have all pointed out that although there was some post-seismic offset following the 1992 Landers and 1999 Hector Mine earthquakes, it does not explain the discrepancy, because a relatively fast geodetic rate [59] had already been observed prior to those two earthquakes.

Another explanation for the discrepancy is the incorporation of off-fault deformation into either the sum geologic rate, the geodetic model used, or both. In this case, much of the shear strain in the ECSZ could be distributed over the entire region and not confined to the fault segments portrayed in models. Analyses that considered off-fault deformation have reduced the discrepancy significantly [21,71–74]. The assumption that some of the fault slip is absorbed by distributed shear (i.e., off-fault deformation) can increase the geologic rate [69]. Similarly, designing the geodetic data inversion model to incorporate off-fault deformation can decrease the geodetic rate [75]. In the end, both approaches minimize the discrepancy, but a difference of a few mm/yr often remains, depending on which values are compared. Therefore, unaccounted-for geologic slip rates from active faults, and strain compatibility with the surrounding region, should be integrated into the sum rate before comparing with geodetic results. The Lavic Lake fault, having one of the largest and most recent surface rupturing events in the ECSZ, is a good candidate to consider, making it a logical starting point for this type of analysis.

## 3. Materials and Methods

### 3.1. Hyperspectral Data Collection

This work was carried out using thermal-infrared (TIR) hyperspectral imagery of the study area. The TIR spectral region is preferred because the major classes of rock-forming minerals exhibit prominent diagnostic spectral features in that band [76,77]. Hence, the hyperspectral TIR approach has proven especially successful in resolving contacts involving carbonate, sulfate, and silicate lithologies [78–81].

Hyperspectral airborne imagery was collected on 27 August 2013 (at 11:00 a.m. Pacific daylight savings time) using Mako, a whiskbroom-type sensor developed by The Aerospace Corporation. Mako measures emitted surface radiance in 128 contiguous TIR channels covering the wavelength range 7.6–13.4 μm [82,83]. The Mako sensor was radiometrically and spectrally calibrated as described in [29,83].

We used a 1.8-km wide, 11-km long swath with 2-m pixel resolution, from a flight at 3660 m (12,000 feet) above ground level (~4570 m or ~15,000 feet above sea level). The footprint of the swath was centered along the Lavic Lake fault and covered the 1999-earthquake maximum slip zone in the Bullion Mountains. In the hyperspectral image data from the Mako whiskbroom sensor, each individual whisk is a data cube, so the words "whisk" and "data cube" are interchangeable. When multiple data cubes are concatenated, the combined set is called a "super cube." The full hyperspectral imagery data set we present consists of two super cubes, with different flight line azimuthal directions to accommodate the change in strike of the Lavic Lake fault's surface trace. When those two super cubes are combined, this is called the "complete image swath."

### 3.2. Ground Truth Field Mapping of the Red Flake Site

We took field photographs and produced a geologic map of the ground truth field mapping area: The Red Flake site (Figures 2–4). Additional details regarding our ground truth field mapping are in [29,84,85].

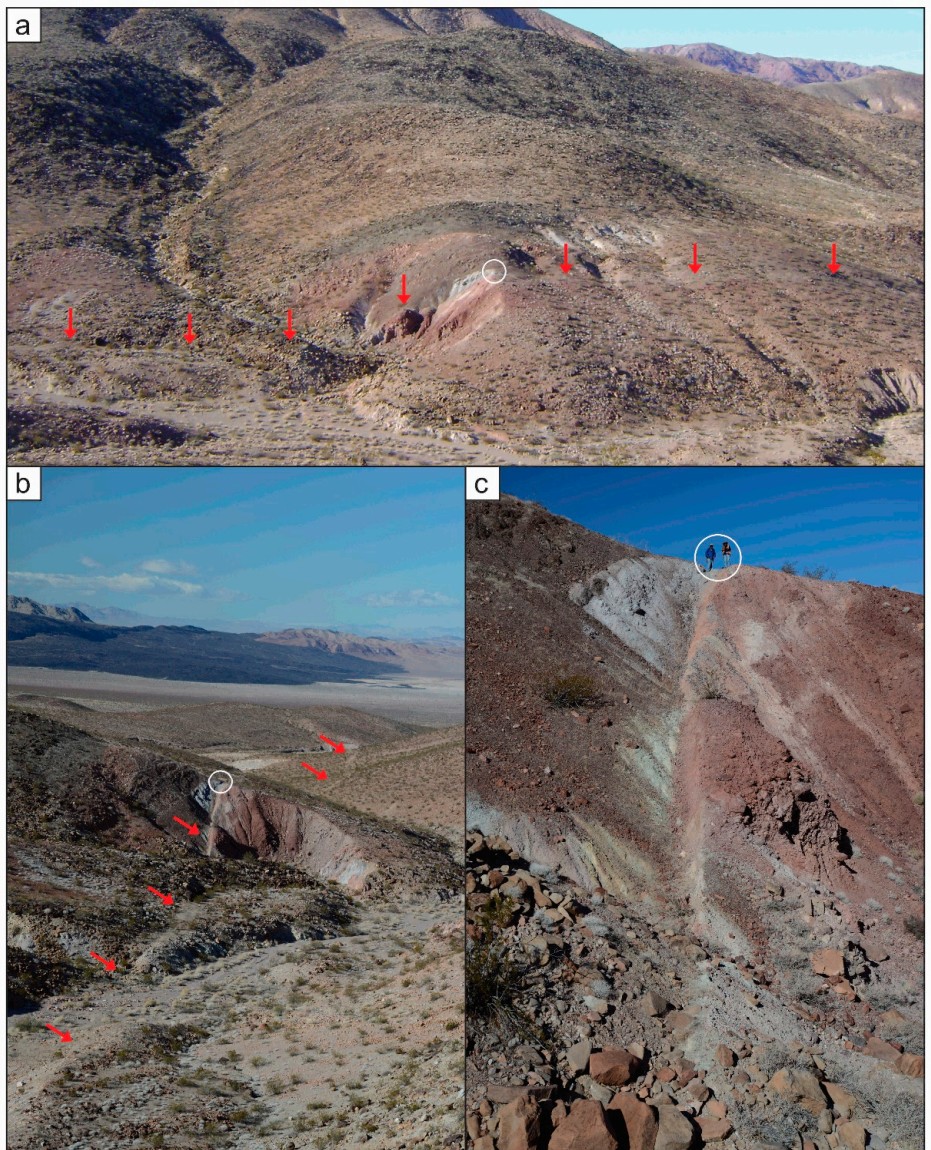

**Figure 3.** Field photographs of the Red Flake site (located at 34°35′10″ north, 116°17′20″ west), a test area where we ground-truthed the lithology as part of the supervised classification process. The true color satellite imagery map of the site is shown in Figure 2f. The lithologic variety at the Red Flake site is discernible by distinct differences in rock color, due to good exposure and little vegetation. (**a**) Overview facing west; arrows highlight the trace of the 1999 primary surface rupture; circle denotes the location where the people seen in (**c**) are standing (photograph date/time/credit: 21 December 2012/9:49 a.m. Pacific standard time/Joann Stock). (**b**) Overview facing north from helicopter; arrows highlight the trace of the 1999 primary surface rupture; circle denotes location where the people seen in (**c**) are standing (photograph date/time/credit: 26 December 2012/1:14 p.m. Pacific standard time/Ken Hudnut). (**c**) View facing north of the 1999 earthquake fault scarp, showing the 1-m-tall protrusion of red feldspar porphyry (center of image, to right of scarp) from which the site derives its name. People on the horizon are circled for scale (photograph date/time/credit: 21 December 2012/10:30 am Pacific standard time/Ken Hudnut).

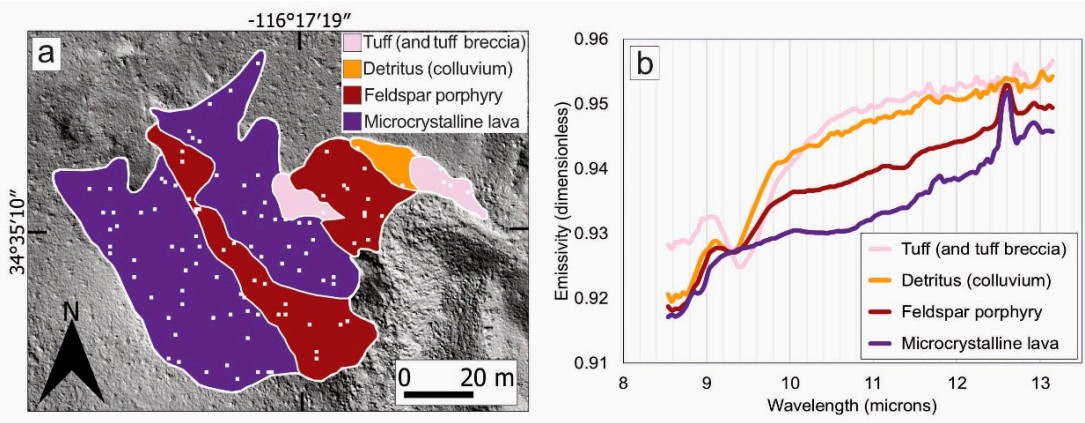

**Figure 4.** (**a**) Geologic map of the Red Flake site, with lithologic names and corresponding color assignments in the spectral plots of (**b**). See [84] for lithologic descriptions. Small white squares are an example of pixels randomly generated to produce end-member spectra for a supervised classification. Random generation is stratified (samples randomly chosen from individual classes) and proportionate to the area defined by a lithologic boundary (we used 2% of the area; e.g., if a certain lithology occupied 100 pixels of image area, two pixels would be randomly chosen for spectral sampling and averaged to define a representative end-member spectrum for said lithology). (**b**) Example remote sensing spectra are apparent emissivity (not directly inferred but calculated from at-sensor radiance, see Figure 5) derived from the randomly generated pixel selections shown in (**a**).

### 3.3. Supervised and Unsupervised Classifications of the Airborne Hyperspectral Imagery

The data processing flow chart shows the steps we used to create the supervised classifications of the Red Flake site test area, and the unsupervised classification of the complete swath of airborne hyperspectral imagery (left and right side paths in Figure 5, respectively). More detailed information about the image processing methods we used, and the image classifications for geologic mapping that we performed can be found in [29,84,85], and Appendix A.1: 'Background on Supervised Classifications and How They Were Applied to This Study.' The minimum noise fraction (MNF) components that were used for the unsupervised classification are displayed in false color assignments (Figure 6) to highlight the lithologic compositional variability across the complete image swath.

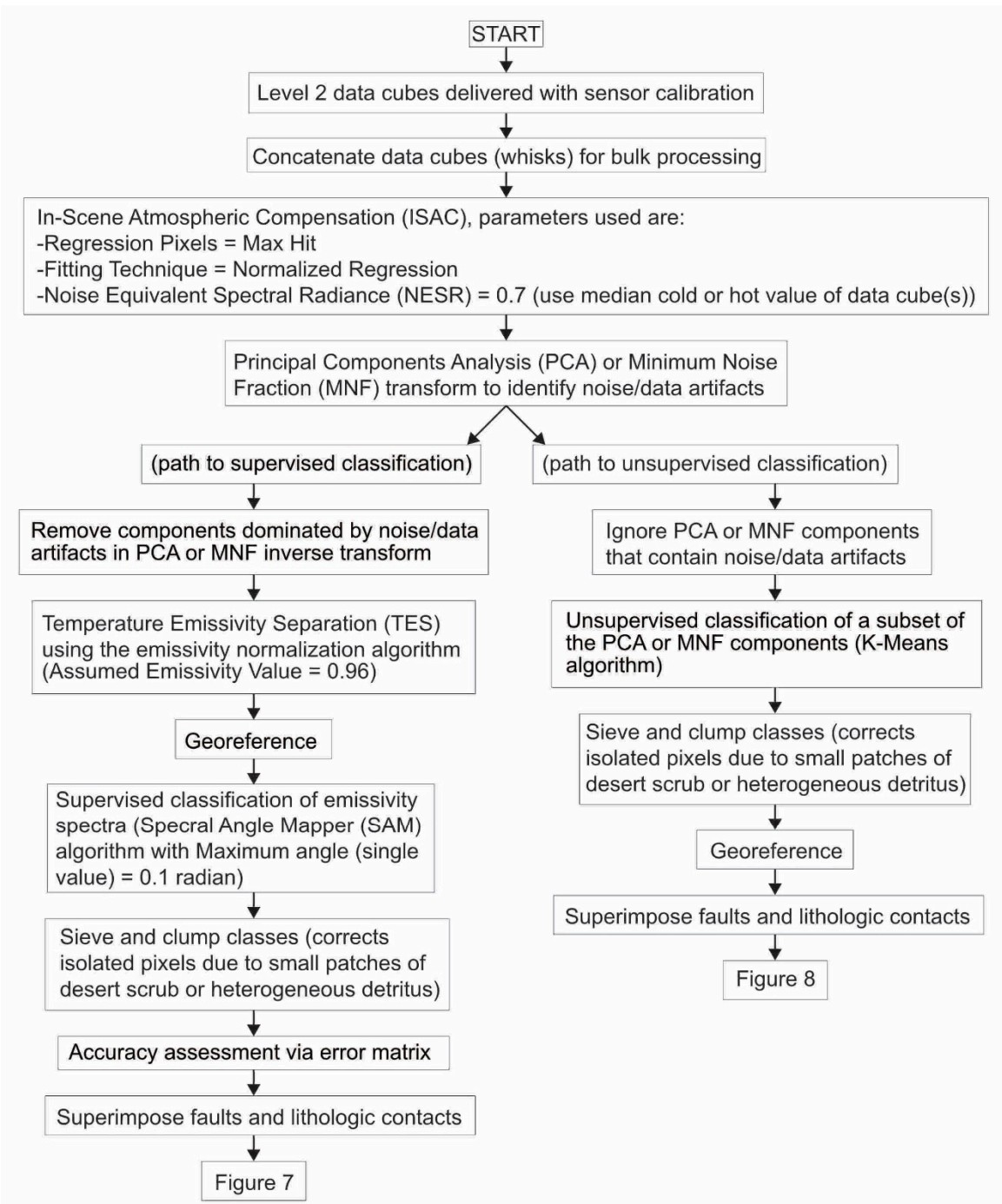

**Figure 5.** Data processing flowchart for our hyperspectral airborne image classifications. We used the Environment for Visualizing Images (ENVI) Software, version 4.8 (Harris Geospatial Solutions, Broomfield, Colorado), and chose standard classification techniques that are commonly available. Level 2 data cubes have undergone radiometric and wavelength calibration, bad pixel replacement, and spectral smile removal (see main text, and [82,83]). User-specified parameters are indicated by how they are entered into ENVI dialogue boxes for each of the image processing steps. References: ISAC [86]; MNF [87]; TES [88]; K-Means [89]; SAM [90]; error matrix [91].

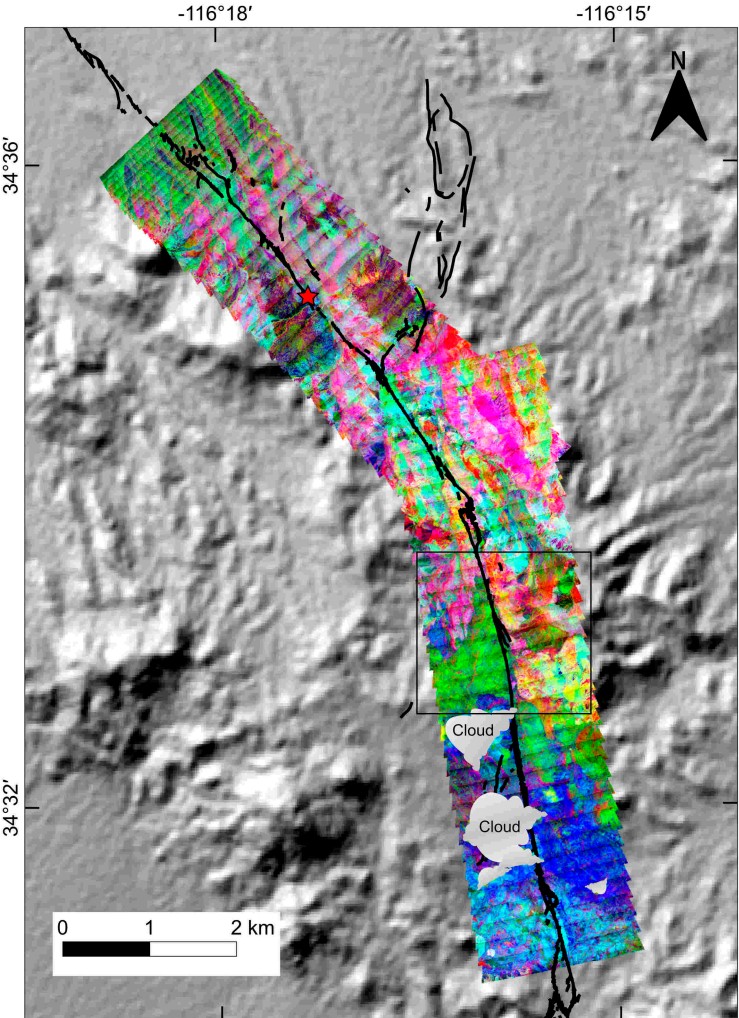

**Figure 6.** Overview of the complete hyperspectral airborne image swath, displayed in the following false color assignment of minimum noise fraction (MNF) components: Red is the third MNF component; green is the sixth MNF component; and blue is the eighth MNF component. MNF components are derived from a modified principal components analysis, where components are ordered by decreasing signal to noise ratio [87]. In this false color assignment, the MNF components represent generalized lithologic variation throughout the scene, where the variation in color is continuous, as opposed to the discretized colors in the classification maps of Figures 7 and 8. A subtle along-track gradation is present, and some clouds that were present during acquisition have been masked and labeled. The Red Flake site is annotated by a star. Rectangular outline is the area shown in more detail in Figures 8–10. GIS data are same as in Figure 1. Map was produced using [12].

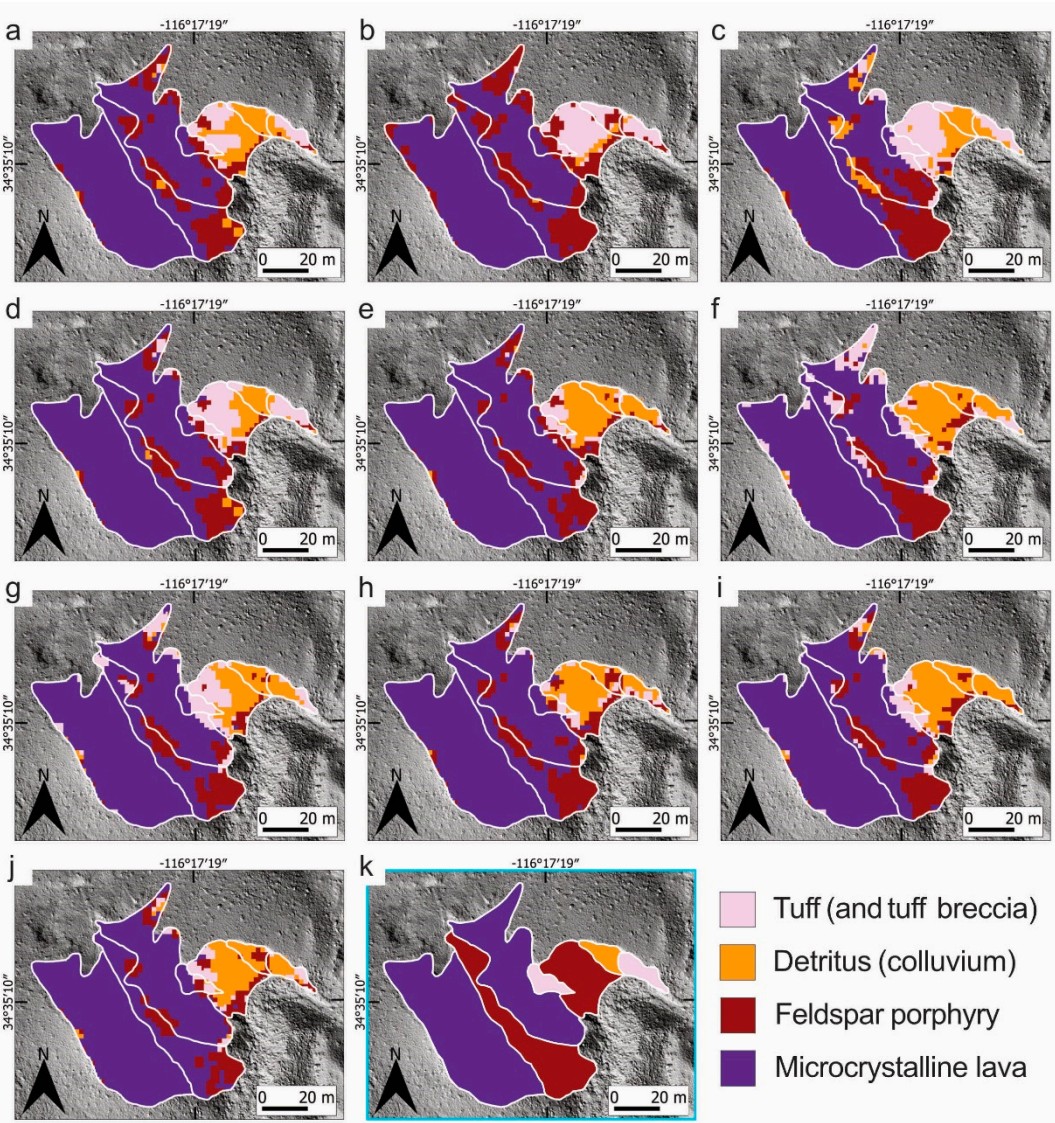

**Figure 7.** (**a**–**j**) Ten supervised classifications of the Red Flake site. Lithologic contacts are superimposed for comparison with the (**k**) ground-truth geologic map. Note that the lithologic ground-truth units here do not necessarily correlate with either the lithologic units from the unsupervised classification (Figure 8), or the geologic maps of [15–19] that are referenced and discussed in the main text. Quantified results for the producer's, user's, and overall accuracies are in Tables 2 and 3, designated by the corresponding letters (**a**–**j**).

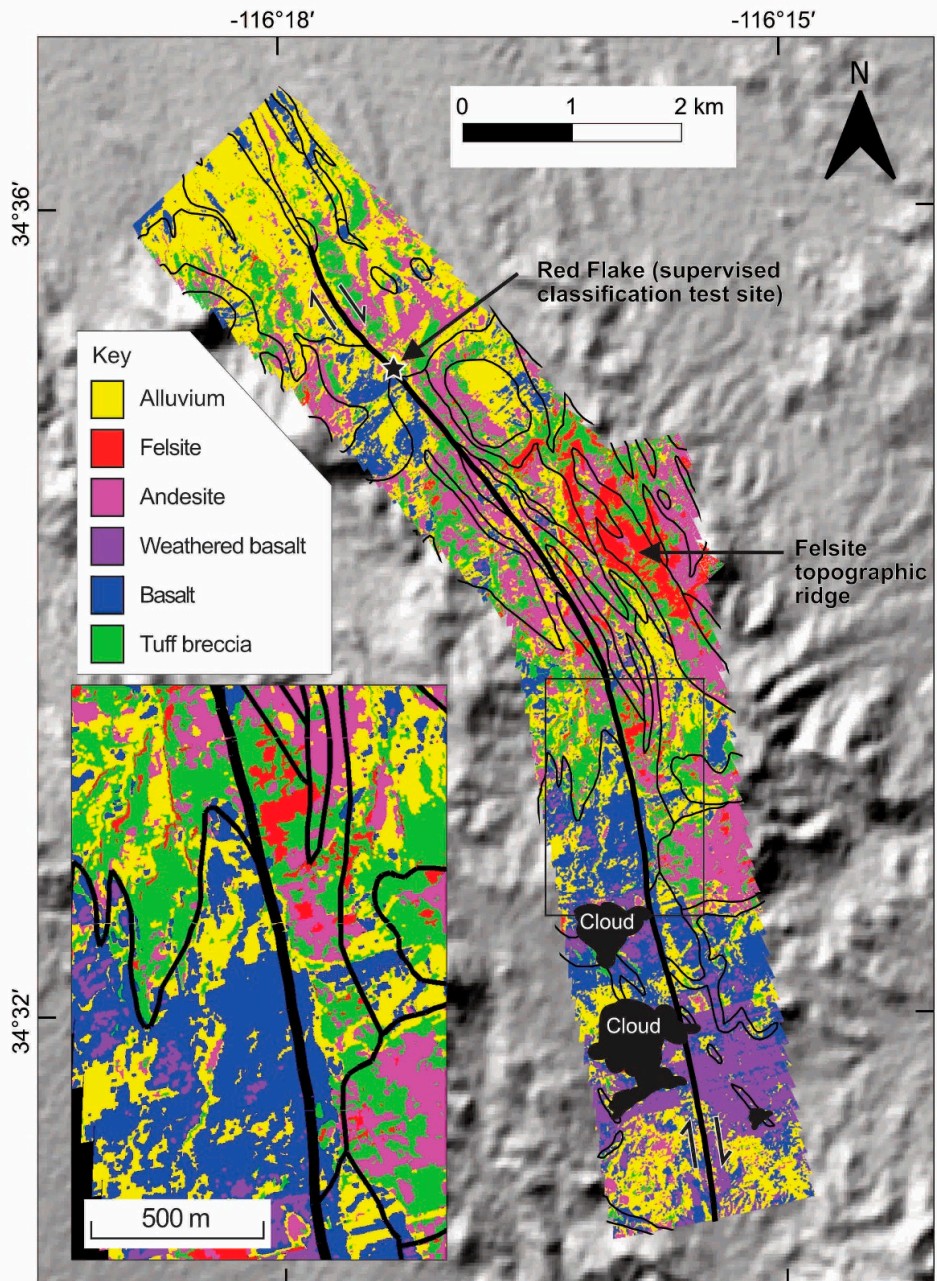

**Figure 8.** Geologic swath map of the Lavic Lake fault from an unsupervised classification of thermal infrared hyperspectral airborne imagery. All line work shown here is from [15], as a layer superimposed on the classification map (thin lines are lithologic contacts, thick line is the approximate surface trace of the Lavic Lake fault). The reader should interpret this map as a comparison between two methods for identifying lithologic contact lines: (1) Lithologic contact lines that were hand drawn in a previous map, and (2) lithologic contact lines defined by the unsupervised classification, which are the boundaries between different colors (i.e., see where the hand drawn contact lines do or do not align with boundaries between different colors). Opposing arrows indicate relative fault motion. Lithologic names shown in key are from correlating our classes with the units from [15] (Table 4). The Red Flake site is annotated by a star. Rectangular outline is the area shown in the zoom inset (lower left, approximate center at 34°33′ north, 116°16′ west), which shows a boundary between our classes that correlates well with a lithologic contact between tuff breccia and basalt. Clouds that were present during image acquisition have been masked and labeled. © 2016 IEEE. Reprinted/modified, with permission, from [29].

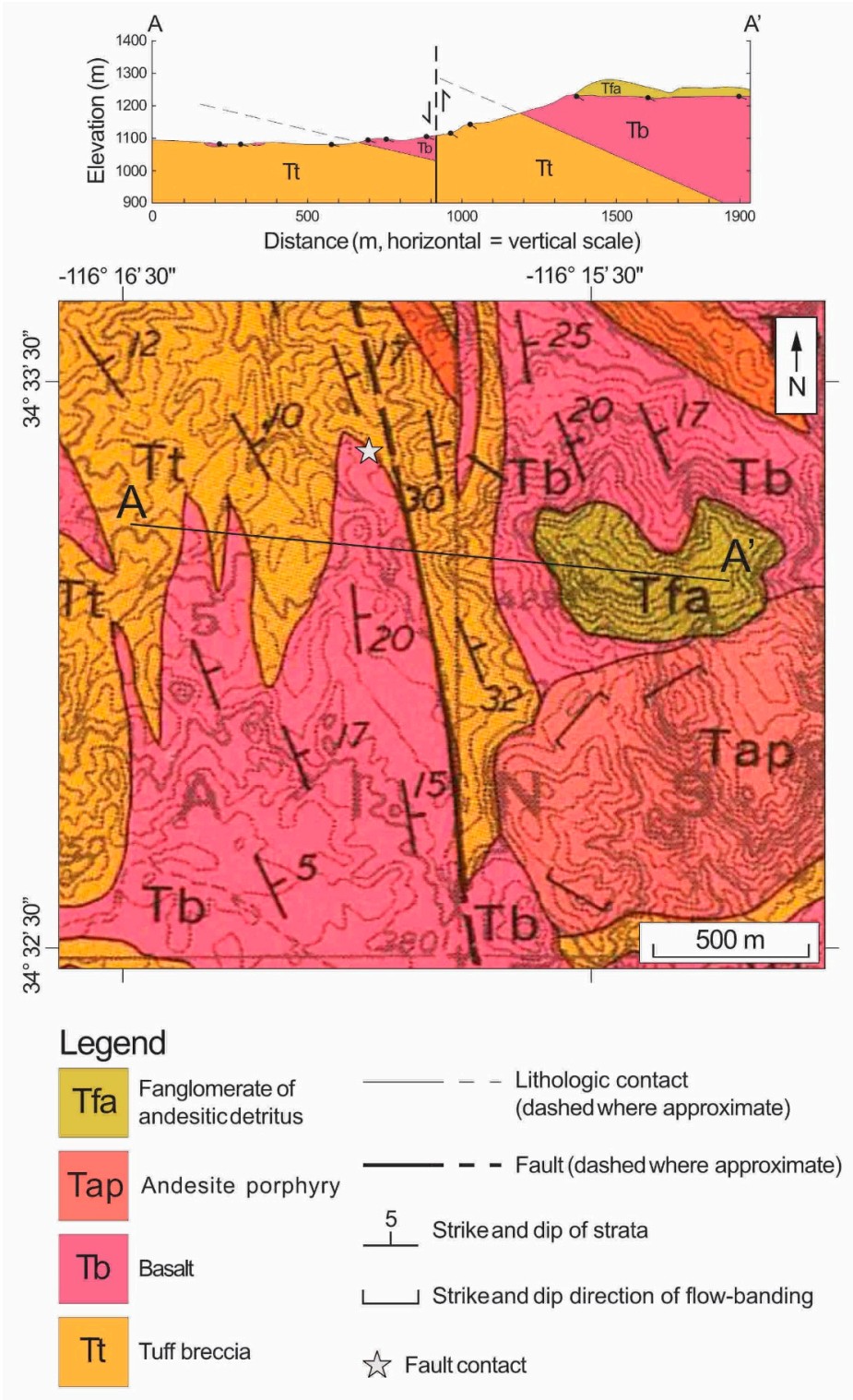

**Figure 9.** Portion of the Dibblee [15] geologic map, zoomed to the same area shown in the zoom inset of Figure 8. The map was modified by adding the line A-A' (with the representative cross section shown above the map), and also adding a star where the Tt/Tb contact was depicted as depositional, but is actually a fault. The legend contains the relevant lithologic units and symbols featured here and discussed in the text. In the cross section, the dip of the Tt/Tb depositional contact is from the average of cross-section-projected dip domains on the west (13° northeast) and east (23° northeast) sides of the Lavic Lake fault. The units are Tertiary (Oligocene or Miocene) in age.

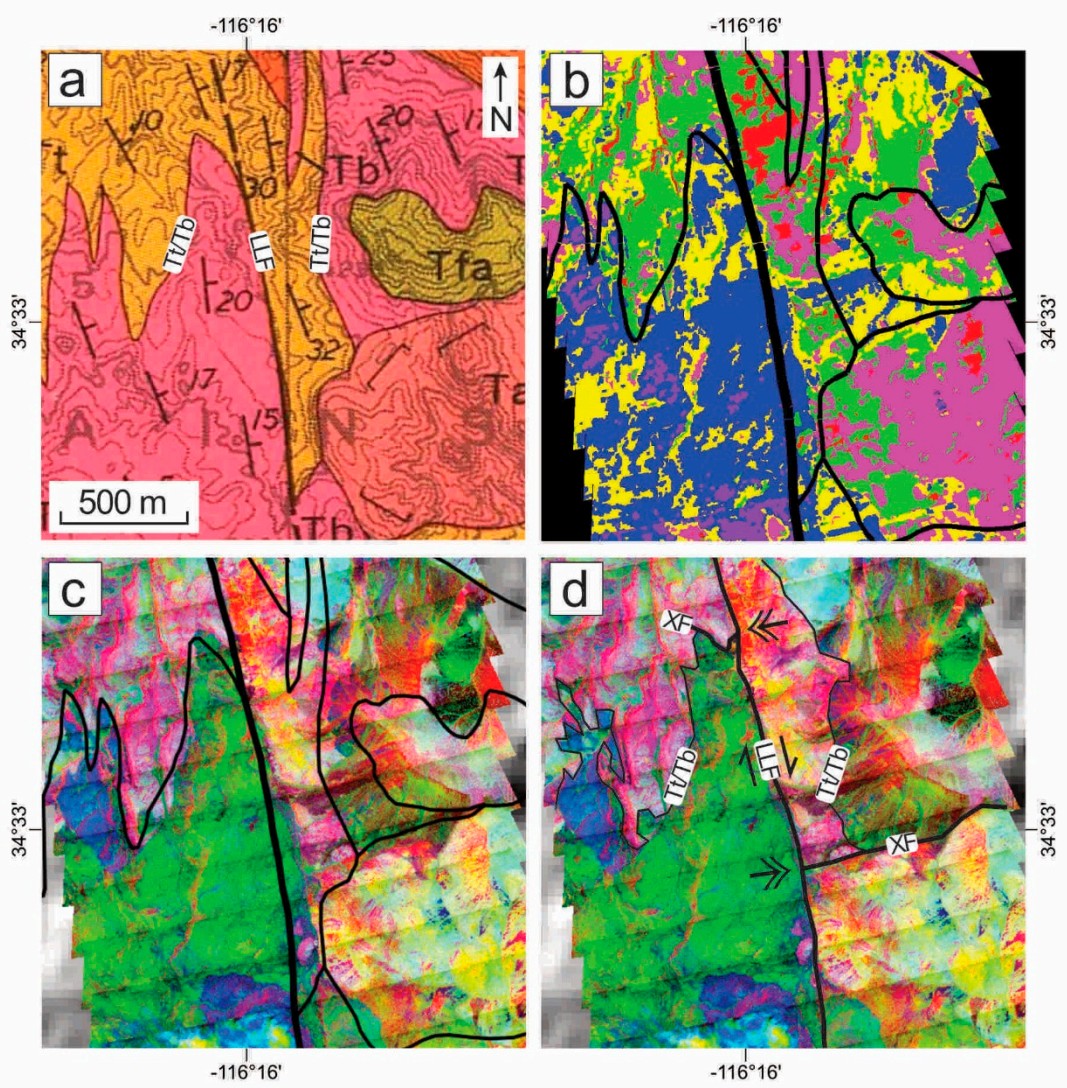

**Figure 10.** Comparison of the cumulative fault offset analysis area as portrayed in the different remote sensing imagery products and maps used in this study. (**a**) the Dibblee [15] map (Figure 9), with the Tt/Tb contact (Tt/Tb) and Lavic Lake fault (LLF) labeled. Scale bar and north arrow in (**a**) apply to all panels. (**b**) the unsupervised classification (Figure 8), with Dibblee's line work superimposed. (**c**) the MNF components in false color (Figure 6), with Dibblee's line work superimposed. (**d**) the MNF components in false color, with new line work annotated to accurately depict the Tt/Tb contact (Tt/Tb), the Lavic Lake fault (LLF), and the cross fault (XF) that is cut and displaced by the main Lavic Lake fault. Opposing arrows indicate the right lateral tectonic motion along the Lavic Lake fault, and the double-headed arrows mark the points where the cross fault intersects the Lavic Lake fault.

**Table 2.** Red Flake site supervised classification error matrices, including producer's, user's, and overall accuracies (see the appendix for description of how to interpret accuracy values). Letters (**a**–**j**) correspond with the classification maps shown in Figure 7 panels (**a**–**j**).

| Classified | Ground Truth | | | | Total | User's accuracy |
|---|---|---|---|---|---|---|
| (a) | Tuff | Detritus | Feldspar porphyry | Microcrystalline lava | | |
| Tuff | 25 | 2 | 56 | 4 | 87 | 0.29 |
| Detritus | 11 | 28 | 69 | 3 | 111 | 0.25 |
| Feldspar porphyry | 13 | 0 | 125 | 89 | 227 | 0.55 |
| Microcrystalline lava | 13 | 0 | 112 | 747 | 872 | 0.86 |
| Total possible | 62 | 30 | 362 | 843 | 1297 | |
| Producer's accuracy | 0.40 | 0.93 | 0.35 | 0.89 | | Overall accuracy = 0.71 |
| (b) | | | | | | |
| Tuff | 30 | 17 | 85 | 2 | 134 | 0.22 |
| Detritus | 2 | 3 | 12 | 0 | 17 | 0.18 |
| Feldspar porphyry | 18 | 10 | 161 | 121 | 310 | 0.52 |
| Microcrystalline lava | 12 | 0 | 105 | 719 | 836 | 0.86 |
| Total possible | 62 | 30 | 363 | 842 | 1297 | |
| Producer's accuracy (%) | 0.48 | 0.10 | 0.44 | 0.85 | | Overall accuracy = 0.70 |
| (c) | | | | | | |
| Tuff | 41 | 6 | 97 | 24 | 168 | 0.24 |
| Detritus | 13 | 23 | 53 | 22 | 111 | 0.21 |
| Feldspar porphyry | 0 | 0 | 128 | 88 | 216 | 0.59 |
| Microcrystalline lava | 8 | 0 | 85 | 709 | 802 | 0.88 |
| Total possible | 62 | 29 | 363 | 843 | 1297 | |
| Producer's accuracy (%) | 0.66 | 0.79 | 0.35 | 0.84 | | Overall accuracy = 0.69 |
| (d) | | | | | | |
| Tuff | 36 | 5 | 81 | 13 | 135 | 0.27 |
| Detritus | 4 | 24 | 43 | 1 | 72 | 0.33 |
| Feldspar porphyry | 9 | 0 | 114 | 68 | 191 | 0.60 |
| Microcrystalline lava | 13 | 0 | 124 | 761 | 898 | 0.85 |
| Total possible | 62 | 29 | 362 | 843 | 1296 | |
| Producer's accuracy (%) | 0.58 | 0.83 | 0.31 | 0.90 | | Overall accuracy = 0.72 |
| (e) | | | | | | |
| Tuff | 13 | 2 | 26 | 7 | 48 | 0.27 |
| Detritus | 29 | 24 | 87 | 2 | 142 | 0.17 |
| Feldspar porphyry | 11 | 3 | 117 | 59 | 190 | 0.62 |
| Microcrystalline lava | 8 | 0 | 132 | 775 | 915 | 0.85 |
| Total possible | 61 | 29 | 362 | 843 | 1295 | |
| Producer's accuracy (%) | 0.21 | 0.83 | 0.32 | 0.92 | | Overall accuracy = 0.72 |
| (f) | | | | | | |
| Tuff | 19 | 1 | 44 | 61 | 125 | 0.15 |
| Detritus | 28 | 24 | 103 | 5 | 160 | 0.15 |
| Feldspar porphyry | 1 | 5 | 124 | 45 | 175 | 0.71 |
| Microcrystalline lava | 14 | 0 | 92 | 731 | 837 | 0.87 |
| Total possible | 62 | 30 | 363 | 842 | 1297 | |
| Producer's accuracy (%) | 0.31 | 0.80 | 0.34 | 0.87 | | Overall accuracy = 0.69 |
| (g) | | | | | | |
| Tuff | 28 | 1 | 70 | 36 | 135 | 0.21 |
| Detritus | 23 | 26 | 68 | 5 | 122 | 0.21 |
| Feldspar porphyry | 1 | 3 | 109 | 45 | 158 | 0.69 |
| Microcrystalline lava | 10 | 0 | 116 | 756 | 882 | 0.86 |
| Total possible | 62 | 30 | 363 | 842 | 1297 | |
| Producer's accuracy (%) | 0.45 | 0.87 | 0.30 | 0.90 | | Overall accuracy = 0.71 |
| (h) | | | | | | |
| Tuff | 15 | 4 | 20 | 7 | 46 | 0.33 |
| Detritus | 22 | 16 | 86 | 3 | 127 | 0.13 |
| Feldspar porphyry | 10 | 10 | 133 | 63 | 216 | 0.62 |
| Microcrystalline lava | 15 | 0 | 124 | 769 | 908 | 0.85 |
| Total possible | 62 | 30 | 363 | 842 | 1297 | |
| Producer's accuracy (%) | 0.24 | 0.53 | 0.37 | 0.91 | | Overall accuracy = 0.72 |

**Table 2.** *Cont.*

| Classified | Ground Truth | | | | Total | User's accuracy |
|---|---|---|---|---|---|---|
| (i) | | | | | | |
| Tuff | 15 | 1 | 27 | 22 | 65 | 0.23 |
| Detritus | 34 | 23 | 98 | 11 | 166 | 0.14 |
| Feldspar porphyry | 4 | 6 | 116 | 55 | 181 | 0.64 |
| Microcrystalline lava | 8 | 0 | 122 | 755 | 885 | 0.85 |
| Total possible | 61 | 30 | 363 | 843 | 1297 | |
| Producer's accuracy (%) | 0.25 | 0.77 | 0.32 | 0.90 | | Overall accuracy = 0.70 |
| (j) | | | | | | |
| Tuff | 16 | 1 | 26 | 10 | 53 | 0.30 |
| Detritus | 22 | 20 | 83 | 6 | 131 | 0.15 |
| Feldspar porphyry | 9 | 8 | 101 | 37 | 155 | 0.65 |
| Microcrystalline lava | 15 | 0 | 152 | 790 | 957 | 0.83 |
| Total possible | 62 | 29 | 362 | 843 | 1296 | |
| Producer's accuracy (%) | 0.26 | 0.69 | 0.28 | 0.94 | | Overall accuracy = 0.72 |

**Table 3.** Red Flake site supervised classification accuracy summary [a]. Accuracy statistics shown here were calculated from values in Table 2.

| Lithology | # of 4-m$^2$ Pixels | Area (m$^2$) | Producer's Accuracy (%) | User's Accuracy (%) |
|---|---|---|---|---|
| Tuff (and tuff breccia) | 62 | 248 | 38 ± 16 | 25 ± 5 |
| Detritus (colluvium) | 30 | 120 | 71 ± 24 | 19 ± 6 |
| Feldspar porphyry | 364 | 1456 | 34 ± 4 | 62 ± 6 |
| Microcrystalline lava | 845 | 3380 | 89 ± 3 | 86 ± 1 |

[a] Standard deviation = 1σ, n = 10; all values rounded to nearest integer. © 2016 IEEE. Reprinted, with permission, from [29].

**Table 4.** Correlation of unsupervised classification units (Figure 8) with lithologic units from the [15] [a].

| Class Color | Lithologic Name Given Here | Letter Codes and Lithologic Names from Dibblee (1966) [b] | Class also Includes these Units [b] |
|---|---|---|---|
| Yellow | Alluvium | Qa: Alluvium<br>Qf: Fan gravel<br>Qoa: Older alluvium<br>Qof*: Older valley sediments, fanglomerate and gravel | QTr: Rhyolitic felsite, (and all others) |
| Red | Felsite | Tif: Intrusive felsite | Ta, Tt |
| Magenta | Andesite | Ta*: Andesite<br>Tap: Andesite porphyry<br>Tfa: Fanglomerate of andesitic detritus | Qof, QTr, Tif, Tb, Tt |
| Purple | Weathered basalt | Tb*: Basalt<br>Tib: Intrusive basalt | Tt |
| Blue | Basalt | Tb*: Basalt<br>Tib: Intrusive basalt | QTr, Ta, Tt |
| Green | Tuff breccia | Tt: Tuff breccia | Qof, Tif, Ta |

[a] See [15–19] for complete lithologic descriptions; correlations are broad generalizations that do not necessarily cover every possible detail or variation. [b] Q: Quaternary, T: Tertiary; when multiple units appear in the third column, asterisks indicate the predominant correlative unit.

## 4. Results

Each supervised classification result (Figure 7) is based on a unique set of end member spectra, a randomly generated subset of each lithologic ground-truth unit within the geologic map of the Red Flake site. Mean accuracy percentage values and standard deviations (all values rounded to the nearest integer) were calculated using error matrices from the ten supervised classifications. Producer's, user's and overall accuracies are tallied in error matrices and summarized in Tables 2 and 3. The producer's accuracy is defined as the fraction of the total number of pixels that classify correctly when reading down a column of the error matrix. Similarly, the user's accuracy is the fraction of the total number of pixels that classify correctly when reading across a row of the error matrix. For further description, see Appendix A.2: 'Guide for Interpreting Error Matrices.'

A primary conclusion from the supervised classification analysis is that the inevitable presence of detrital material (e.g., colluvium/alluvium/etc.), will lower the accuracy ratings of lithologic classification maps [29,84]. The tuff and tuff breccia (producer's accuracy = 38 ± 16%) are often incorrectly classified as detritus, and the feldspar porphyry (producer's accuracy = 34 ± 4%) is often incorrectly classified as microcrystalline lava. With similar mineral assemblages, the incorrect classifications could be partly due to compositional similarity between lithologies. The user's accuracy of the tuff and tuff breccia is 25 ± 5%, but the feldspar porphyry has a user's accuracy (62 ± 6%) that is nearly double its producer's accuracy. For practical applications, the user's accuracy can be a more important statistic, given that it is the probability that a material, when checked in the field, will actually be what the map claims that it is [91]. The microcrystalline lava had the highest classification ratings for both its producer's and user's accuracies (89 ± 3% and 86 ± 1%, respectively). Overall, our statistics show that out of the four lithologic units that we used as ground truth, the microcrystalline lava is probably the most dependable for image classification mapping.

Our goal with the unsupervised classification analysis was to identify features such as lithologic contact lines that could be used for cumulative offset measurements. We used lithologic contact lines from published geologic maps [15–19] to investigate this further. Hereafter, references to the work of Dibblee include all of those publications, but the primary reference is [15]. We superimposed Dibblee's lithologic contact lines onto our unsupervised classification map (Figure 8) to help us define lithologic compositions for our unsupervised classification units (Table 4), and to visually inspect and compare where lithologic boundaries apparent in the unsupervised classification correlated with the mapped lithologic contact lines.

## 5. Discussion

### *5.1. Displaced Features*

Our analysis is based on observations from the geologic maps, high-resolution satellite imagery, and ground-based field photographs. By discerning boundaries between lithologic units with our new maps, we identified correlative planar features on both sides of the Lavic Lake fault: A lithologic contact, and an older cross fault. The intersection of those planes is a piercing line in older bedrock units that was subsequently cut by the Lavic Lake fault. We measured the net slip as the distance between the points where the linear features pierced the Lavic Lake fault plane.

### 5.1.1. Tt/Tb Contact

The key lithologic units are a tuff breccia and a basalt ("Tt" and "Tb" respectively, Figures 8–10 and Table 4), which are designated and referenced from the combined analysis of our unsupervised classification with Dibblee's geologic map (note that these units do not necessarily correlate with any similar or identically-named units from our supervised classification analysis at the Red Flake site). We determined that bedding in Tt and Tb is right way up, based on attitudes plotted on the Dibblee map for crude bed forms within these units that are apparent in the oblique view satellite imagery and field photographs (Figures 11 and 12, respectively). In true color, Tt is a very light shade of tan,

and Tb is an overlying, very dark shade of brown. Tb is younger than Tt by stratigraphic superposition. The contact between the two units is depositional; this feature is the "Tt/Tb contact."

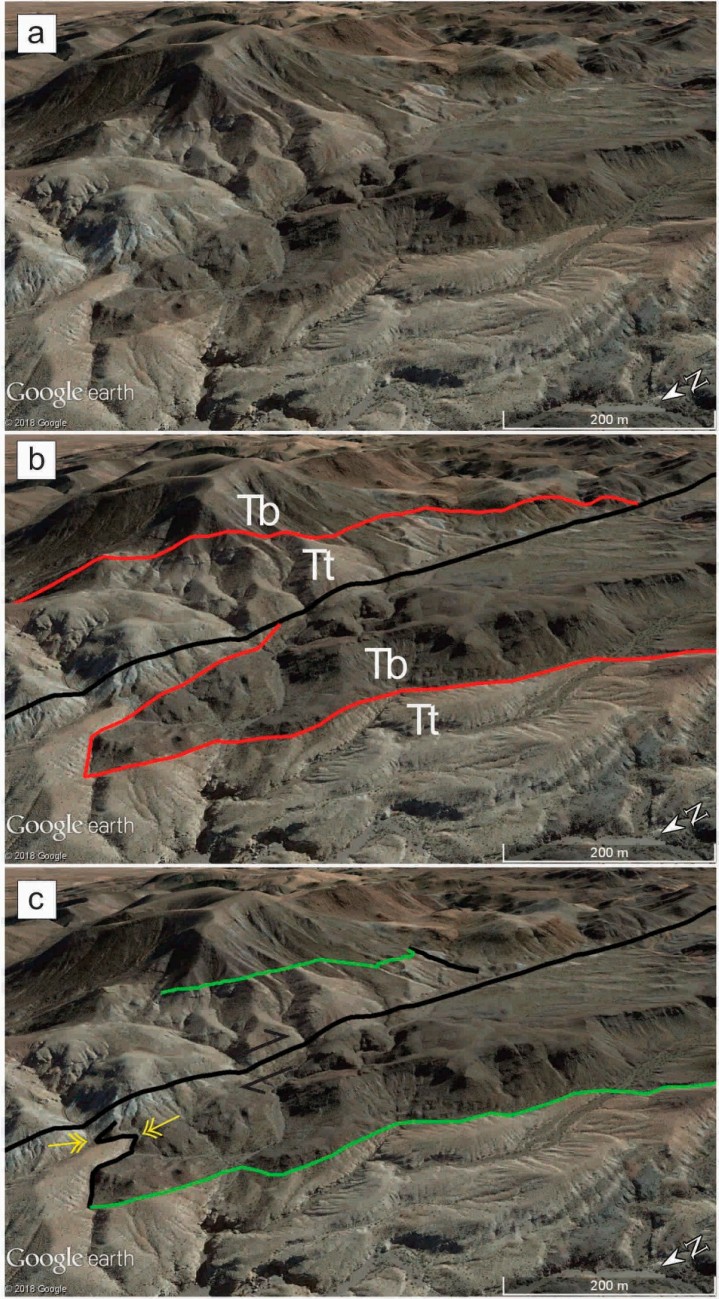

**Figure 11.** Oblique-view Google Earth satellite image (imagery date: 2 January 2015) of the location (34°33′ north, 116°16′ west) where we measured net slip on the Lavic Lake fault. Scales shown are not accurate everywhere in the images due to the oblique viewing geometry. (**a**) Image without annotation. (**b**) Thick black line is the Lavic Lake fault surface trace, thin red lines are Dibblee's Tt/Tb lithologic depositional contact, which separates the lighter-hued lithology (Tt, the older tuff breccia), from the darker-hued lithology (Tb, the younger basalt that overlies Tt). (**c**) Thick black line is the Lavic Lake fault surface trace, thin green lines are the Tt/Tb depositional contact (modified from Dibblee's depiction), and the smaller cross faults are also depicted by thick black lines. Note that the smaller cross fault on the west side of the Lavic Lake fault is depicted in (**b**) as a portion of the Tt/Tb depositional contact. The yellow double-headed arrows are the ground-based field photograph locations in Figure 12.

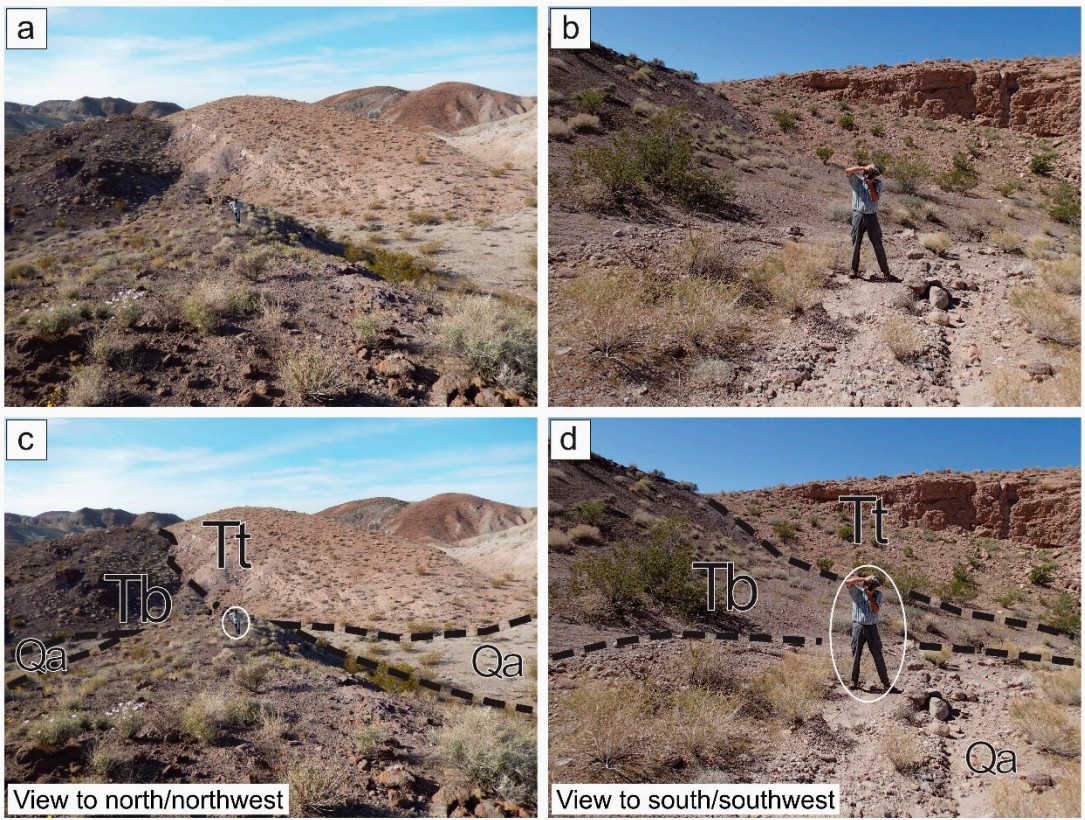

**Figure 12.** Ground-based field photographs of the cross fault (Tt/Tb fault contact) on the west side of the main Lavic Lake fault. Figure 11c shows the photograph locations as yellow double-headed arrows. Person (Ken Hudnut) provides scale. (**a**) date: 3 April 2014; time: 4:09 p.m. Pacific daylight savings time; location: 34.5562° north, 116.2672° west; viewing direction: North/northwest; taken by Joann Stock. (**b**) date: 2 October 2012; time: 10:55 a.m. Pacific daylight savings time; location: 34.5571° north, 116.2670° west; viewing direction: South/southwest; taken by Frank Sousa. (**c**) same as (**a**), but annotated with lithologic letter codes (Table 4), and dashed contact lines. (**d**) same as (**b**), but annotated with lithologic letter codes (Table 4), and dashed contact lines. In (**a**), bedding in Tt can be observed dipping to the east/northeast (to the right side of the photograph). In (**c**,**d**), note that the older Tt unit is structurally above the younger Tb unit.

On the west side of the Lavic Lake fault, the Tt/Tb contact appears in the previously published geologic maps [14,15], and in our unsupervised classification map (Figures 8 and 10b), the Tt/Tb contact is revealed as a generalized boundary between the green-color-coded Tt, and immediately to the southeast, the blue-color-coded Tb. With Dibblee's lithologic contacts superimposed on our unsupervised classification, the distinction between the tuff breccia and basalt becomes more evident (also see MNF components in false color, Figures 6 and 10c). Tt and Tb strata, and their contact boundary, generally dip 15–30° northeast. The strike of Tt/Tb bedding and their contact is fairly consistent in proximity to the Lavic Lake fault, and the very low angle of intersection between the contact trace and the main fault contributes to an exaggerated amount of perceived lateral separation in map view (e.g., Figure 8.6 in [92]; Figure 6.48 in [93]). While the Lavic Lake fault has a net slip that predominantly resulted from right-lateral strike slip, the cross section (line A-A' in Figure 9) shows that the Tt/Tb contact also has apparent vertical separation, revealing a solution that could include some pre-1999, east-side-up cumulative vertical offset of the Tt/Tb contact. The vertical offset is also evidenced by: (1) Topographic relief, with greater surface elevation on the east side of the Lavic Lake fault than on the west side (Figure 9); and (2) the observation from [10] that the vertical component of slip from the 1999 earthquake was also predominantly east-side-up in the Bullion Mountains section.

A younger rhyolitic felsite ("QTr," Table 4) with a relatively flat base overlies the tilted volcanic bedrock units Tt, Tb (and their depositional contact) in angular unconformity. Note that QTr is *not* the same lithology as the felsite class in Figure 8; that felsite was correlated with an intrusive felsite ("Tif," Table 4). A bedrock exposure labeled "QTr" in Dibblee's maps is now known to be Peach Spring Tuff [94], which has implications for age control. The age of the Peach Spring Tuff is 18.78 ± 0.02 Ma [95], which predates the ECSZ age of inception (Table 1) and therefore the Lavic Lake fault. The Tt/Tb contact predated the Peach Spring Tuff, and thus captured the inception and cumulative offset of the Lavic Lake fault.

### 5.1.2. Cross Fault

The Tt/Tb contact was previously depicted as depositional on both sides of the Lavic Lake fault. On the west, the contact line trends along a 5–10° azimuth, but ~200 m from the main fault trace, the contact line makes an abrupt 120° turn before intersecting the main fault trace, which we attribute to a separate, smaller cross fault ("XF" in Figure 10d). The Tt/Tb fault contact here is further evidenced by a structural displacement: Ground-based field photographs show the older, light-hued Tt structurally above the younger, dark-hued Tb (Figure 12). While we interpreted this field relationship as a fault contact, we note the possibility that the Tt and Tb units are in an interfingering contact, as implied by others [14,15].

On the east side of the main fault, the Tt/Tb contact is also depositional, but does not clearly align with a distinct class boundary from our remote sensing map. This is partly because the Tt/Tb contact does not intersect the main fault on its east side; instead, the contact is truncated by a separate, smaller cross fault ("XF" in Figure 10d; and [14]). Figures 10 and 11 compare the previous map with our interpretation of the same features, including the smaller cross faults in relation to the Tt/Tb contact and the main fault. The east and west cross faults could be a correlative structure that predates inception of the main fault. If that is true, then the intersection line of the Tt/Tb contact with the cross fault serves as an offset linear feature from which to measure net slip along the Lavic Lake fault.

The cross fault on the east appears to have a steep dip, based on its relatively straight surface trace, which strikes at high angle to the Lavic Lake fault. The cross fault on the west, however, could have a lower dip angle, as suggested by its surface trace. We recognize that if the cross faults are not correlative, the displacement measurement is not valid. However, the Lavic Lake fault's vertical component of tectonic deformation could have tilted one or both of the displaced cross fault segments into different orientations. One possibility is that if the cross fault originally had a shallow dip, the east block could have experienced tilting during relative uplift by the Lavic Lake fault, creating a steeper dip angle and without requiring a symmetric mode of deformation in the west block.

### *5.2. On-Fault Net Slip and Off-Fault Deformation*

With planar features identified, we used attitude data to calculate the orientation of intersection lines, and then projected those lines in to the main fault to determine the location of the piercing points. The attitude of the Tt/Tb contact is from strike/dip data in the Dibblee map, 341/15 (341 ± 11, 15 ± 4, n = 6) on the west side of the main fault, and 343/25 (343 ± 8, 25 ± 6, n = 5) on the east. We estimated the attitude of the cross fault from our new hyperspectral mapping, 285/45 on the west (strike from two points at equal elevation on the surface trace, dip estimated from field photographs and nature of surface trace), 254/90 on the east (strike and dip from relatively straight surface trace). The trend/plunge of the intersection line is 90/14 (90 +6/−5, 14 ± 5) on the west, and 74/25 (74 ± 0, 25 ± 6) on the east. The on-fault net slip is the distance between the piercing points: 1036 +27/−26 m, which can be decomposed into 1008 +14/−17 m horizontal and 241 +51/−47 m vertical components.

Our net slip analysis only applies to discrete, on-fault deformation along a single structure, and thus does not include the proportion of long-term deformation accommodated away from the main fault. Off-fault deformation has been cited as a potentially significant source of error in considering the discrepancy between long-term geologic and current geodetic slip rates [69,71–75]. A greater amount

of off-fault deformation is likely to occur in unconsolidated sediments (e.g., the Lavic Lake playa to the northwest, or alluvium to the southeast), than in the bedrock where we performed this cumulative slip analysis, but any additional integrated displacement could be substantial.

Milliner et al. [74] used optical image correlation to calculate 39 ± 22% average horizontal off-fault deformation (OFD) at the surface from the 1999 Hector Mine earthquake. The 1999 Hector Mine earthquake OFD is accommodated over an average fault zone width of 121 m for the entire surface rupture, and 16 m for the section that corresponds with our net slip measurement. For their OFD values that were calculated at the same location as our horizontal component of slip, the average value is 21 ± 11% (n = 7). If we assume that the Milliner et al. [74] OFD values apply to the complete displacement history of the Lavic Fault, then 21% horizontal OFD increases the 1008 +14/−17 m horizontal slip component to 1276 +18/−22 m. The entire-surface-rupture statistical maximum OFD value of 39 + 22 = 61% increases our slip to a maximum 2621 m. While the assumption that the 1999-OFD applies to the complete slip history is probably not accurate, some OFD has certainly contributed and therefore must be added. OFD can be more significant on younger faults, as emphasized by the paleoseismology-based interpretation that the Lavic Lake fault is still relatively young [34].

Our OFD-inclusive value of 1276 +18/−22 m is significantly less than the previously reported magnetic gradient offset estimate of 3400 ± 800 m [20], but our 2621-m extreme maximum case does overlap with their minimum. The difference between our surface measurement and their magnetic measurement may be due to several factors: (1) Our measurement location is 3–5 km north of theirs; (2) we projected piercing lines to the Lavic Lake fault from 10–100 m distance, whereas they projected from >1 km away (points A′ and A″ in Figure 6 from [20]); and (3) 3D basement geometry, which is not resolvable in this analysis.

### 5.3. Implications for Estimates of Slip Rate on the Lavic Lake Fault

The Lavic Lake fault would need to have a 4-mm/yr slip rate to bring the ~6-mm/yr sum ECSZ geologic rate (since ~750 ka [22]) within the 10–15 mm/yr range of contemporary geodetic rates [59–65]. With a hypothetical 4-mm/yr slip rate, and the 1276 m slip from this study, the inception age of the Lavic Lake fault would be 319 ka. If that age is accurate, the Lavic Lake fault is a relatively young structure in in the >5-Ma ECSZ architecture (Table 1). A young Lavic Lake fault agrees with the paleoseismic interpretation [34], with a separate, proximal strand rupturing not in 1999 but sometime within the past ~1750 years. If earthquakes occurred once every 1750 years for 319,000 years, each event with 5 m of slip like the 1999 earthquake, only 911 m of slip would accumulate. A 5-m-slip earthquake every 1750 years would take 447,000 years to accumulate 1276 m slip, and more conservatively, a 2.5-m-slip earthquake every 1750 years would take 893,000 years. These calculations support a >319 ka age for the Lavic Lake fault, which implies a geologic slip rate of <4 mm/yr.

If tectonic motion along the Lavic Lake fault initiated at 750 ka, the geologic slip rate with 1276-m cumulative slip would be 1.7 mm/yr. The true slip rate could be between 1.7 and 4 mm/yr, but anywhere in that range is too low to completely reconcile the difference between ECSZ sum geologic and geodetic slip rates. The Lavic Lake fault appears to be one of the few known major faults whose geologic slip rate is unknown and has not yet been integrated into the sum geologic rate. However, if more faults are discovered in the future, it still might be possible to resolve the discrepancy by integrating more geologic slip rates from currently unknown faults.

Geologic slip rates are often minimum values due to uncertainty in lag time between the age of the displaced feature and inception of fault motion. Assuming this error source manifests consistently, that in itself could explain the discrepancy, because the true geologic rate would be larger than any value calculated with available methods. On the other hand, if the geologic rates are accurate and the discrepancy really does exist, perhaps the ECSZ is currently experiencing some type of true transient or permanent accelerated deformation rate.

### 6. Conclusions

The key results and conclusions of this study are:

1. Hyperspectral-airborne-image geologic maps of a test site along the 1999 Hector Mine earthquake surface rupture are accurate to 71 ± 1%
2. The net slip along the Lavic Lake fault is 1036 +27/−26 m, which increases the horizontal component to 1276 +18/−22 m by incorporating off-fault deformation
3. The estimated long-term slip rate is <4 mm/yr, which does not raise the sum geologic ECSZ rate to present-day geodetic values

We used thermal infrared hyperspectral airborne imagery with remote sensing methods, field work, published geologic maps, and satellite imagery to identify displaced features and measure net slip along the Lavic Lake fault. Some of the class boundaries in our remote sensing image products display compelling correlation with lithologic contacts that were previously identified. A mapped lithologic contact between a tuff breccia (Tt) and a basalt (Tb) correlates well with a boundary between two of the classes from our unsupervised classification geologic map. At >18.78 ± 0.02 Ma, the Tt/Tb contact predates ECSZ inception at 5-10 Ma, and therefore also the Lavic Lake fault.

The Tt/Tb contact intersects with a cross fault to form a linear feature that is displaced 1036 +27/−26 m by the main Lavic Lake fault. The on-fault net slip is only a fraction of the total, with horizontal OFD from the 1999 earthquake extrapolated to increase the cumulative long-term offset of bedrock recorded at the surface to 1276 +18/−22 m. The maximum possible value we derived for cumulative offset is 2621 m, which agrees in the error range of a 3400 ± 800-m displaced magnetic feature measurement. The difference between our surface measurement and the magnetic measurement could be due to different analysis locations, distances from which we projected our piercing lines in to the fault, and unknown 3D basement geometry.

The net slip that we calculated can assist in reconstructing fault histories in eastern California, and can also be combined with bedrock ages to calculate the geologic fault slip rate. Based on the available constraints, our Lavic Lake fault slip rate estimate of <4 mm/yr does not sufficiently increase the sum geologic Mojave ECSZ slip rate to geodetic values. Our new datum instead supports the interpretation that the rate discrepancy is real, which could mean that the ECSZ is currently experiencing a transient or permanent accelerated deformation rate. This rate change could explain the increasing frequency of large earthquakes in eastern California over the past few decades, beginning in the 1990s with Landers and Hector Mine, and continuing with the 2019 Ridgecrest events.

**Author Contributions:** Conceptualization, R.A.W., J.M.S. and D.M.T.; methodology, R.A.W., J.M.S., D.M.T., K.N.B., D.K.L. and F.J.S.; software, R.A.W., J.M.S., K.N.B. and P.D.J.; validation, R.A.W., J.M.S, D.M.T., K.N.B., P.M.A., P.D.J.; formal analysis, R.A.W.; investigation, R.A.W., J.M.S., D.M.T, K.N.B., P.M.A., P.D.J., D.K.L., F.J.S.; resources, R.A.W., J.M.S., D.M.T., P.M.A.; data curation, R.A.W., J.M.S., D.M.T. and K.N.B.; writing—original draft preparation, R.A.W. and J.M.S.; writing—review and editing, R.A.W., J.M.S., D.M.T., K.N.B., P.M.A., P.D.J., D.K.L. and F.J.S.; visualization, R.A.W., P.D.J. and F.J.S.; funding acquisition, R.A.W, J.M.S. and D.M.T. All authors have read and agreed to the published version of the manuscript.

**Funding:** This research was funded by the National Science Foundation (NSF) Graduate Research Fellowship Program under Grant No. 1144469 awarded to R. Witkosky, and by Southern California Earthquake Center (SCEC) Award No. 14160 awarded to J. Stock. This paper is SCEC Contribution No. 8898. SCEC is funded by NSF Cooperative Agreements EAR-1033462 & EAR-0529922, and United States Geological Survey Cooperative Agreements G12AC20038 & 07HQAG0008. Mako airborne hyperspectral imagery was acquired under the auspices of the Aerospace Corporation's Independent Research and Development program.

**Acknowledgments:** We thank the Marine Corps Air Ground Combat Center in Twentynine Palms, California, for allowing access to the military base. We thank Ken Hudnut, Janet Harvey, Kate Scharer, and Sinan Akçiz for their help and support in fieldwork, performing this research, and preparing this manuscript. We also thank the Editors for handling this manuscript, and Nick Van Buer and two anonymous reviewers for helping improve the content. The new data presented in this study are available through Caltech's Research Data Repository (https://doi.org/10.22002/d1.1182). The data used in this study from previously published geologic maps are available from the U.S. Geological Survey's National Geologic Map Database (https://ngmdb.usgs.gov/ngmdb/ngmdb_home.html); the references also indicate the specific URLs for each item.

**Conflicts of Interest:** The authors declare no conflict of interest.

**Appendix A**

*Appendix A.1. Background on Supervised Classifications and How They Were Applied to This Study*

After compiling our ground-truth data into a geologic map, we used this map data to perform supervised classifications on the portion of the hyperspectral imagery that covers the Red Flake site. A supervised classification is a common remote sensing product in which pixels in an image are organized into a set of classes defined a priori by a user who has knowledge of materials or land cover present in the image. In defining the classes, a few pixels are chosen to represent each class as "end members." Then, each other pixel in the image is grouped with the end member that has the greatest spectral similarity. For our case, the classes are the four main lithologic units that we observed in the field.

After pixels are organized into classes, physical boundaries between classes are superimposed on the image in order to determine how many pixels in each area were assigned to their correct class. Results for correctly and incorrectly identified pixels are tabulated and offer a quantitative summary (as the percentage of pixels mapped correctly in relation to ground truth) for the accuracy to which the classes can be mapped using a remote sensing image. For our purpose, the supervised classification was a test to establish how well spectral information from the thermal infrared hyperspectral airborne imagery allowed for differentiation of distinct lithologic units on the scale of the pixel size.

For further clarity, we present an example case: a lithologic unit is observed and its boundary (contact line) is mapped in the field. A remote sensing image completely covers the map view extent of this unit, so the lithologic boundary is digitized and superimposed on the image. Spectra from a few pixels within the boundary are chosen to represent the lithologic unit. In an ideal supervised classification, all of the other pixels within the boundary should be grouped, based on spectral similarity, with the chosen representative pixels. In practice, though, complications exist that make a 100% accuracy rating for a supervised classification highly unlikely (see Discussion section in main text; [29,84]).

The spectral angle mapping algorithm (SAM, described in [90]) calculates the n-dimensional angle, where n is the number of bands, between the spectrum for each pixel and the spectrum for each end member using the geometric definition of an inner product. Each pixel is then classified as the end member for which the minimum angle (greatest spectra similarity) is calculated. Finally, lithologic contacts from our geologic map were superimposed to quantify (and assist in visualizing) the classification accuracy, and we assessed the accuracy of the supervised classification via error matrices [91]. Figure 7 shows the results for the ten supervised classifications (labeled 'a' through 'j'), and the corresponding error matrices are in Table 2.

*Appendix A.2. Guide for Interpreting Error Matrices*

Here, we provide a detailed explanation, with examples, for how to interpret error matrices and calculate producer's, user's, and overall accuracies. All of the following explanations reference numerical values in the error matrix from supervised classification 'a' in Figure 7 and Table 2.

Reading down a column of the error matrix shows the distribution of classified pixels within each ground truth class. For example, the first column shows that in the bounded region that we mapped as tuff, 25 pixels were correctly classified as tuff, but 11, 13, and 13 pixels were incorrectly classified as detritus, feldspar porphyry, and microcrystalline lava, respectively. Thus, summing over a column gives the total number of pixels contained in each bounded ground truth area. The total number of pixels classified correctly as a fraction of the total number of pixels in a column gives the "Producer's accuracy". The name is in reference to a scenario where the producer of the classification map wishes to assign a grade to their product, so they simply quantify how many pixels are correct within each bounded class area. For the tuff, this is 25/62 = 0.40 or 40%.

Reading across a row of the error matrix shows how many pixels for each type of lithology were classified into each ground truth class. For example, the first row shows that 25 true tuff pixels were correctly classified as tuff, but 2 true detritus, 56 true feldspar porphyry, and 4 true microcrystalline lava pixels were incorrectly classified as tuff. Thus, summing over a row gives the total number of pixels classified as a given lithology throughout the entire classification map. The total number of pixels classified correctly as a fraction of the total number of pixels in a row gives the "User's accuracy." The name is in reference to a scenario where a user of the classification map field-checks every pixel and then assigns a grade based on misclassified pixels across the entire scene, not just grading within the individual class boundaries. For the tuff, this is 25/87 = 0.29 or 29%.

Taking the sum of the diagonal of the error matrix gives the total number of pixels classified correctly in relation to ground truth. The total number of pixels classified correctly across all classes, as a fraction of the total number of pixels in the classification map gives the "Overall accuracy," which is the lowest entry to the right. For supervised classification "aA" and its accompanying error matrix, this is 25 + 28 + 125 + 747 = 925, then 925/1297 = 0.71 or 71%.

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
