# Peer review of "The Lavic Lake Fault: A Long-Term Cumulative Slip Analysis via Combined Field Work and Thermal Infrared Hyperspectral Airborne Remote Sensing"

_remotesensing, doi:10.3390/rs12213586_

Round 1

Reviewer 1 Report

I found this to be an interesting, well written, and well illustrated paper.  I think this paper is a great model for how remote sensing should be used, as it goes in some depth into both the remote sensing methods, and also, crucially, their geologic interpretation.  The soundness of the classification method is well demonstrated by the ground-truth field mapping comparison, and should prove to be another useful tool for geologic mappers looking to leverage remotely sensed data in inaccessible areas.

I do have a few minor comments/concerns (see below), but I think they will not be too difficult to address:

Line 285: Have you evaluated the hypothesis that the low user’s accuracy for colluvium may partly just reflect the differing scales of field and hyperspectral mapping? I mean, since field geologists normally generalize out thin or small patches of colluvium if they can reasonably infer what’s underneath, but this sort of classification addresses each pixel separately (other than a bit of sieve and clump). On the other hand, this seems to have been mapped at a pretty detailed scale, so maybe that’s not what's going on here . . . .

Line 426: The use of a single cross fault that may or may not actually correlate across the Lavic Lake Fault is one thing (I appreciate that you include some caveats), but the apparent dip diffferences in the cross faults on either side of the Lavic Lake Fault would appear to suggest rotation around a roughly horizontal, east-west axis. A rotation in this direction 1) seems like an odd orientation for folding related to the Lavic Lake fault, and 2) would also rotate the bedding, which is not observed.

A more attractive interpretation, based, I will admit, solely on my analysis via Google Earth, might be that the N-S lineament between your yellow arrows in Fig 11c is, in fact, simply part of an unmapped splay of the Lavic Lake Fault itself, with a bit of negative flower structure action (along a releasing bend, no less) lowering Tb on its east side. In this case, the west-side cross fault may actually be steeply dipping also. The strike would then be NW, which still seems a little off; but, in this case, this sense of vertical axis rotation is at least in an unsurprising orientation for a drag fold related to the Lavic Lake fault. (And if this is a drag fold related to the Lavic Lake fault, you might also be able to constrain a bit of off-fault deformation, too . . .)

Line 444: Don’t you want just the strike-slip component for the following analysis, not the net slip??  It would also be good to quantify the vertical component to reinforce the points made about dip slip earlier, around line 397.

Line 487: This statement seems a bit hubristic and unfounded, given the number of sizeable earthquakes that keep happenning on previously unrecognized faults (or faults of previously unrecognized significance). It seems likely enough to me that there could be plenty of unrecognized small- to medium-sized faults in the ECSZ to make up this discrepancy . . . . If you’re going to make this statement, you probably at least want to cite someone . . . .

But all in all, I like this paper a lot, and I look forward to its publication!

Reviewer 2 Report

This paper uses hyperspectral LWIR airborne imagery to accurately map lithological contacts in (relatively) well-exposed, structurally-complex igneous extrusive rocks in order to improve estimation of fault movements.   The paper is clear and well written.   Figures are useful and generally appropriate.  

Several specific comments:

  • It is not clear what value the classification results add to the task of mapping lithological contacts (e.g. Tb/Tt)  especially when a simple colour composite of selected MNF bands (derived from ISAC processed data) appears to yield the best result (Figure 10d). Removing the classification work (and related TES) would better focus and shorten this paper considerably. 
  • The MAKO pixel/ROI “emissivity” spectra presented in Figure 4b do not look realistic in that they all show a ramp-up towards longer wavelengths. Do you have field/lab spectra for comparison/validation.  I also note the wavelength start at ~8.5 microns which is too long for targeting the silicate Christiansen Frequency for intermediate silicate rocks.   Why didi you start from this wavelength and not ~7.8 microns.  Peak at ~12.7 microns for microcrystalline lava is also weird.   Any comments on its cause (it is not a silicate particle size effect)?
  • Is it correct to say that from Figure 11, the same Tb/Tt contact seen in the MAKO MNF product (Figure 10d) is also evident in natural colour imagery (available in GoogleEarth™)?  If so then why bother with the LWIR for your task, especially if you can use the publish geology to get you into the right lithological ball-park using free satellite VNIR imagery.

Reviewer 3 Report

Dear Editor and Authors,

Let me start by saying that I’m not an expert on hyperspectral imaging. I have read with interest the Material and Methods section of this paper and I can only say that it is well designed for non expert readers, like me. I enjoyed the integration of field and remotely sensed data to reduce uncertainties in geological mapping and I think that this paper represents a nice case study, suitable for publication.

The text is well written and illustrations are clear. I found that some improvements could be easily made. Concerning the more geological aspect of the paper, I’m sceptical about the existence of the eastern transverse fault and I think that the authors should either provide more convincing data or adopt a different structural model, see my major comment below.

Major point.

The cumulative displacement along the Lavic Lake Fault is computed using two transverse faults affecting the eastern and western block, respectively. The authors interpret them as a formerly integer fault that has been displaced by the Lavic L Fault. While the eastern transverse fault is evident, the western one is not.

I take a tour in Google Earth of the area shown with the two arrows in fig 11c (I have used historical imagery; 3/2010 is the best), see the attached pdf:

The Tb-Tt contact (yellow arrows) is slightly discordant on Tt (layering in Tt is indicated by violet arrows). In the area where the authors infer a fault, the contact between the two formations is sharp and highly oblique to layering in Tt, which I agree may suggest the occurrence of a normal fault. However, if you follows the best fit plane of the contact (which strikes NNW-SSE) you can see that layers in Tt (violet arrows) are not displaced. A very similar situation can be observed to the west (upper inset in the pdf), suggesting the occurrence of some kind of topography before the deposition of Tb.

Minor points.

Apart from the comment above (which concerns the discussion), the only section I would rearrange is the geological setting. I can easily follow it because I’m a structural geologist. I think that, due to the audience of the journal, the authors could try to make it shorter and simpler. Not all the information provided there is strictly needed in this work.

Line 17 tectonic reconstruction and risk assessment

L 22. 71% with respect to what?

L 23 1035 +24/-27 ?

L 72-75 is the slip rate over geological time known or not? it is (or seems) unclear

L 106 co-seismic displacement

L 107 composition of these rocks

L 111 major structure in the area

Figure 2a miss “fault” for the Buillion fault

A geological map of the area of figure 2a could help

L 124 several important results.

Report the results in form of a list, i.e. 1)….2)…

Figure 4a. Add a legend for the different units

Figure 7 I would add a different border (width and colour) to 7k, to differentiate it from the other maps

Figure 9 add age of units

Figure 12 add orientation to photos
